# Synthesis of Hyaluronic Acid-Conjugated Fe_3_O_4_@CeO_2_ Composite Nanoparticles for a Target-Oriented Multifunctional Drug Delivery System

**DOI:** 10.3390/mi12091018

**Published:** 2021-08-26

**Authors:** Chang Ryong Lee, Gun Gyun Kim, Sung Bum Park, Sang Wook Kim

**Affiliations:** 1Department of Advanced Materials Chemistry, Dongguk University, Gyeongju 38066, Korea; no4623@dongguk.ac.kr (C.R.L.); gungyun88@dongguk.ac.kr (G.G.K.); 2Department of Safety Engineering, Dongguk University, Gyeongju 38066, Korea; parksungbum@dongguk.ac.kr

**Keywords:** superparamagnetic iron oxide, cerium oxide, hyaluronic acid, nanoparticles, drug delivery system

## Abstract

This study is based on the principle that superparamagnetic iron oxide nanoparticles (Fe_3_O_4_) can be used to target a specific area given that their magnetic properties emerge when an external magnetic field is applied. Cerium oxide (CeO_2_), which causes oxidative stress by generating reactive oxygen species (ROS) in the environment of tumor cells, was synthesized on the surface of superparamagnetic iron oxide nanoparticles to produce nanoparticles that selectively kill cancer cells. In addition, hyaluronic acid (HA) was coated on the cerium’s surface to target CD44-overexpressing tumor cells, and ^nat^Zr was chelated on the Fe_3_O_4_@CeO_2_ surface to show the usefulness of labeling the radioisotope ^89^Zr (T1/2 = 3.3 d). The synthesis of Fe_3_O_4_@CeO_2_ was confirmed by Fourier Transform-Infrared Spectroscopy (FT-IR), X-ray Diffraction (XRD) and Field Emission-Transmission Electron Microscope (FE-TEM). The coating of HA was confirmed by FT-IR, X-ray Photoelectron. Spectroscopy (XPS), FE-TEM, Energy-Dispersive X-ray Spectroscopy (EDS) and Thermogravimetric Analysis (TGA)/Differential Scanning Calorimetry (DSC). The sizes of the prepared nanoparticles were confirmed through FE-TEM and Field Emission-Scanning Electron (FE-SEM) (sizes of 15 to 30 nm), and it was confirmed that ^nat^Zr was introduced onto the surface of the nanoparticles using EDS. The particle size of the dispersed material was limited through Dynamic Light Scattering (DLS) to about 148 nm in aqueous solution, which was suitable for the (enhanced permeation and retention) EPR effect. It was confirmed that the HA-coated nanoparticles have good dispersibility. Finally, a cytotoxicity evaluation confirmed the ability of CeO_2_ to generate ROS and target the delivery of HA. In conclusion, Fe_3_O_4_@CeO_2_ can effectively inhibit cancer cells through the activity of cerium oxide in the body when synthesized in nano-sized superparamagnetic coral iron that has magnetic properties. Subsequently, by labeling the radioactive isotope ^89^Zr, it is possible to create a theranostic drug delivery system that can be used for cancer diagnosis.

## 1. Introduction

Drug delivery systems are a new and rapidly developing science, wherein substances on the nanoscale are used as diagnostic tools or to deliver therapeutics to specific target sites in a controlled manner [1,2,3]. When large substances are used for drug delivery, problems arise, such as instability in the body, reduced bioavailability, reduced solubility, reduced absorption capacity, reduced target-specific delivery and the possibility of side effects [4,5]. However, nanoscale materials have many theoretical advantages, including high loading capacity, specific targeting and sustained release control [6,7,8]. In addition, intravenously administered nanoparticles of macromolecular anticancer agents tend to circulate for long periods of time, unless they are small enough to be excreted by the kidneys or large enough to be rapidly recognized and captured by the reticuloendothelial system (RES) [9]. A major limitation of most existing chemotherapeutic agents is their lack of tumor selectivity. As such, passive targeting and active targeting are employed, using tumor characteristics to direct nanoparticles towards cancer. Tumors accumulate rapidly, forming new blood vessels with many defects, and the cells inside these blood vessels become weak due to their abnormal vascular structure. As a result, nanoparticles that are tens to hundreds of nanometers in size can enter cancerous tissue more easily; the immune system surrounding the tumor cells collapses and the lymphatic vessels cannot function normally. As such, the nanoparticles containing the drug are retained in the tumor long enough to break down and release the drug. This phenomenon is called the EPR (enhanced permeation and retention) effect. Nano-sized anticancer drugs show excellent therapeutic effects while minimizing side effects by selectively injecting anticancer substances at high concentrations into tumor tissues via the EPR effect [10,11]. In addition, a targeting ligand that can selectively bind to a specific antigen or receptor overexpressed on the surface of the new blood vessels or cancer cells can be effectively targeted for nanoparticle binding, which is called active targeting. Additionally, some structural considerations are important when designing nanoparticles for medical applications: they should comprise of a surface that is biologically inert, be stable under physiological conditions, be capable of moving freely through the body, safely encapsulate chemicals and be readily conjugated to the desired targeting antibody [12].

When superparamagnetic iron oxide is synthesized with lipids or polymers on its surface, its stability is increased, it can be chelated using various ligands and it can be drawn into cancer cells with a magnet. Iron oxide nanoparticles have previously been synthesized via methods such as all milling or co-precipitation, and with the recent developments in synthesis technology related to the pyrolysis of metal precursors, nanoparticles of various sizes (ranging from 1 to 100 nanometers) can be uniformly synthesized. Uniformly synthesized nanoparticles are very important for two reasons. First, the nanoparticles injected into a human body have different biodistributions according to their size. Nanoparticles measuring 10 nm or larger can stay in the blood for a relatively long time, and when there are blood vessels with wide gaps, such as in the case of cancer, they can be easily delivered to the relevant tissue. Therefore, the uniformity of nanoparticles is important from a medical point of view. Second, the magnetic properties of iron oxide nanoparticles also vary with size. Superparamagnetic iron oxide in its bulk state displays strong ferrimagnetism, even in the absence of an external magnetic field. However, when the oxide is nano-sized, the thermal energy outside the particle at room temperature becomes greater than the magnetic energy inside the particle. In this state, in the absence of an external magnetic field, the magnetic moment of the superparamagnetic iron oxide nanoparticle becomes random, due to the thermal energy, resulting in the absence of magnetism. However, when a magnetic field is applied from the outside, the magnetic moments of the particles are arranged in a certain direction by the magnetic field, thereby exhibiting strong magnetic properties. This property is called superparamagnetism, and it is a unique characteristic of superparamagnetic iron oxide nanoparticles. Paramagnetism, however, is a weaker magnetic state than superparamagnetism, while ferromagnetism has a higher magnetic field strength; however, in this case, residual magnetism remains even when the external magnetic field is removed, meaning nanoparticles can aggregate in the blood vessels. With all this in mind, superparamagnetic iron oxide nanoparticles are suitable for medical use because they maintain a stable colloidal form since there is almost no magnetic interaction between the particles when no external magnetic field is present [13,14].

During oxidative stress, the quantities of free radicals in cells increase in an unbalanced way through generation and decomposition, and when the free radicals in cells increase, deoxyribonucleic acid (DNA) and lipids can be damaged [15]. As such, cerium nanoparticles exploit the abnormal redox state of cancer cells to induce cancer cell death through oxidative stress, which has no side effects on normal cells. In particular, since two oxide (Ce^3+^/Ce^4+^) states coexist on the surface of cerium oxide, the redistribution of charge gives rise to oxygen vacancies in the nanostructured lattice, which can cause a reversible redox reaction. This treatment selectively induces cancer cell death through oxidative stress and effectively saves normal cells at the same time [16]. Due to the loss in redox regulation, more reactive oxygen species (ROS) are produced in cancer cells compared to normal cells [17]. In addition, intracellular ROS accumulation inhibits cell circulation in the G1 or G2/M phase, thus disrupting cell proliferation. Therefore, cerium oxide nanoparticles have potential utility as cancer therapeutics because the regulation of ROS can inhibit the growth of cancer [18].

Hyaluronic acid (HA) is a glycosaminoglycan with a structure of repeating disaccharides of d-glucuronic acid and N-acetyl d-glucosamine units, and it is a biosynthetic natural substance. It is also used as an anticancer agent because it is biocompatible, biodegradable and nontoxic. Hyaluronic acid targets the CD44 receptor, which is widely expressed in various tumor types, such as ovarian, breast, colon, stomach and acute leukemia, as well as in cancer cells, and it can target cancer cells via ligand–receptor action [19,20]. HA is synthesized by HA synthase (HAS1 and HAS2) on the inner surface of the cell membrane and is secreted into the extracellular matrix (ECM). The high-molecular-weight HA present in the ECM binds to the CD44 receptor on the cell surface. It is then degraded inside the cell via hyaluronidases (Hyal-1 and Hyal-2). Hyal-1 is a major enzyme mainly found in tumor tissues [21,22]. The receptor-mediated internalization of hyaluronic acid catabolism first binds to the receptors (CD44) on the surface of cancer cells and is then degraded to 50 saccharide units by Hyal-2 on the cell’s surface to form caveolae. These caveolae become endosomes and finally fuse with lysosomes, at which point, they are further broken down into tetrasaccharides by Hyal-1 [22,23]. Therefore, the presence of hyaluronic acid on the surface of nanoparticles makes it possible to target tumors.

In radiopharmaceuticals, radioactive isotopes are labeled on drugs and administered to the human body to diagnose or treat diseases. About 80% of total radiopharmaceuticals are used for diagnosis and 20% for treatment. Positron tomography (PET) is a test method that can display images of the human body in three dimensions using radiopharmaceuticals that emit positrons. It is a useful test for diagnosing various cancer types, evaluating recurrence and confirming treatment. ^89^Zr has a relatively low positron energy of 395.5 keV and a half-life of 78.4 h, which makes it safer to manage and dispose of in clinics, easier to produce and more stable in vivo. As such, using it for high-resolution, quantitative imaging through PET is much more effective than using ^124^I to treat tumors [24,25,26]. When radionuclides are released, ^89^Zr can accumulate in bones or bind to plasma proteins, so the release of ^89^Zr must be prevented. Appropriate chelators are needed to minimize its separation from the drug. The following chelators have been identified: 1,4,7,10-tetraacetic acid (DOTA), ethylenediaminetetraacetic acid (EDTA), Diethylenetriaminepentaacetic acid (DTPA) and desferoxamine (DFO). However, Zr-DOTA, Zr-DTPA and Zr-EDTA are unstable. However, the chelator of Zr and DFO has shown good stability, less than 0.2% of the Zr^4+^ is released from the serum after 24 h, and no more is released within 7 days [25,27,28].

In this study, by synthesizing core–shell composite nanoparticles that have a superparamagnetic core and a ceria shell, an anticancer agent that induces cancer cell death through oxidative stress was prepared using the abnormal redox state of cancer cells. In addition, by coating them with hyaluronic acid, it was possible to target the CD44 receptor, which is widely expressed in cancer cells, thereby enhancing the effect of anticancer drugs. Finally, Zr was coordinated on the surface of the nanoparticles using DFO. Therefore, the possibility of introducing ^89^Zr, a radioactive isotope for PET, for use in the future as a theranostic anticancer drug that can be used for treatment and diagnosis was confirmed.

## 2. Materials and Methods

### 2.1. Materials

Iron (II) sulfate heptahydrate and iron (III) chloride hexahydrate from Daejung (Seoul, Korea); sodium hydroxide, triethylamine (TEA), (3-Aminopropyl)triethoxysilane (APTES), cerium (III) nitrate hexahydrate, hyaluronic acid sodium salt from Streptococcus equi (HA), 3-(4,5-Dimethylthiazol-2-yl)-2,5-Diphenyltetrazolium Bromide (MTT) and zirconium (IV) chloride from Sigma-Aldrich (St. Louis, MO, USA); N-hydroxysuccinimide (NHS) and 1-(3-dimethylaminopropyl)-3-ethylcarbodiimide hydrochloride (EDC) from TCI (Tokyo, Japan); HEPES buffer solution, Roswell Park Memorial Institute (RPMI) Medium 1640, Dulbecco’s Modified Eagle Medium (DMEM), Penicillin-streptomycin, fetal bovine serum (FBS), Phosphate-buffered saline (PBS) and antibiotic-antimycotic (Anti-Anti) from Gibco (Grand Island, NE, USA); KB, CT-26 and MDA-MB-231 from Korean Cell Line Bank (Seoul, Korea); p-NCS-Bz DFO from CheMatech (Dijon, France); dimethyl sulfoxide (DMSO) from Kanto Chemical (Tokyo, Japan); X-ray Diffraction (XRD) from Rigaku (Tokyo, Japan); Fourier transform infrared spectroscopy (FT-IR) from Bruker (Billerica, MA, USA); X-ray photoelectron spectroscopy (XPS) from Thermo Fisher (Waltham, MA, USA); Dynamic light scattering (Malvern, UK); field emission transmission electron microscope (SEM); field emission-scanning electron microscope (FE-SEM) and energy-dispersive X-ray spectroscopy (EDS) (Hitachi, Tokyo, Japan); Thermogravimetric analysis (TGA) and differential scanning calorimetry (DSC) from TA (New Castle, DE, USA).

### 2.2. Methods

#### 2.2.1. Preparation of Superparamagnetic Iron Oxide

The superparamagnetic iron oxide was prepared by co-precipitation. Iron (II) sulfate heptahydrate and iron (III) chloride hexahydrate were pulverized and dissolved in distilled water at a molar ratio (1:2) and then further dissolved using an ultrasonic cleaner for 30 min. Then, the dissolved solution was placed drop by drop into 100 mL of 0.9 M sodium hydroxide in the absence of oxygen using a dropwise funnel. The solution was stirred at 80 °C for 1 h, then washed it 3 times in distilled water and 3 times in ethanol (Figure 1).

#### 2.2.2. Synthesis of Fe_3_O_4_@CeO_2_

The prepared superparamagnetic iron oxide nanoparticles were dispersed in 50 mL of distilled water, and cerium (III) nitrate hexahydrate was completely dissolved in 50 mL of distilled water. The two solutions were then mixed. At this time, the mass ratio of the solute was 1:10. The mixed solution was placed drop by drop into 100 mL of sodium hydroxide (0.9 M) in the absence of oxygen. This was then stirred at 90 °C for 24 h and washed with distilled water 3 times and ethanol 3 times. Finally, the calcinate was placed in a furnace at 300 °C for 1 h (Figure 1).

#### 2.2.3. Synthesis of Fe_3_O_4_@CeO_2_-APTES

We dispersed the prepared Fe_3_O_4_@CeO_2_ in ethanol, added 2 mL of APTES (4.27 M) in a vial at room temperature, stirred it for 12 h and washed the solution 3 times with ethanol (Figure 1).

#### 2.2.4. Synthesis of Fe_3_O_4_@CeO_2_-APTES-DFO

We dispersed 10 mg of Fe_3_O_4_@CeO_2_-APTES in dimethyl sulfoxide, added 1 mg of p-NCS-Bz DFO (1.33 µmol), then stirred the mixture at room temperature for 1 h in a vial and washed it 3 times with distilled water (Figure 1).

#### 2.2.5. Synthesis of Fe_3_O_4_@CeO_2_-APTES-DFO-Zr

We dispersed 10 mg of Fe_3_O_4_@CeO_2_—APTES-DFO in 3 mL of distilled water in a vial; then, 1 mL of HCl (0.1 M) and 0.4 mL of HEPES (1 M) solution were added. Next, 1 mg of zirconium (IV) chloride (1.33 µmol) was added, and the mixture was stirred at room temperature for 1 h then washed three times with distilled water (Figure 1).

#### 2.2.6. Coating of HA on the Surface of Fe_3_O_4_@CeO_2_-APTES-DFO-Zr

We dispersed Fe_3_O_4_@CeO_2_-APTES-DFO-Zr in 50 mL of distilled water in a 500 mL round floor flask. Then, we added 370 mg NHS (3.22 mmol), 200 mg EDC (1.04 mmol) (using the EDC/NHS activation approach, the amine group, which is the terminal group of APTES, and the carboxyl group of HA were combined) and 113 mg HA (0.77 µmol). After stirring at 60 °C for 30 min with complete dissolution, we mixed the two solutions and added 2 mL of TEA (7.13 M) (TEA acts as a buffer). After stirring at room temperature for 24 h, we washed the mixture with distilled water 3 times and ethanol 3 times, then dried it in a vacuum (Figure 1).

#### 2.2.7. Cell Viability Assay

KB, CT-26 and MDA-MB-231 (CD44 overexpressing human breast cancer cells [29]) cells were plated using 96-well plates at a cell density of 1 × 10^5^ cells per well (5 × 10^4^ for CT-26) in 100 μL of cell culture medium. KB and CT-26 were incubated in DMEM (10% FBS, 1% Anti-Anti) and MDA-MB-231 in RPMI (10% FBS, 1% Penicillin) for 24 h (at 37 °C and 5% CO_2_). After that, the experiment was carried out. A total of 100 mL PBS with MTT (0.5 mg/mL) was put into a well, then dissolved in DMSO after 3 h, and absorbance at 570 nm was measured.

## 3. Results and Discussion

### 3.1. Fourier Transform Infrared Spectroscopy (FT-IR) and X-ray Photoelectron Spectroscopy (XPS)

FT-IR analysis was performed for calcined samples in the range of 4000–500 cm^−1^ to determine the surface modification of the catalyst (Figure 2). For Fe_3_O_4_ and Fe_3_O_4_@CeO_2_, the broad vibration peak at 3392 cm^−1^ is associated with O-H stretching, while that at 1626 cm^−1^ is associated with bending. The surfaces of the two materials made of Fe_3_O_4_ and Fe_3_O_4_@CeO_2_ are clearly hydrophobic, and so the materials are suitable for surface modification. In Fe_3_O_4_@CeO_2_-APTES, the absorption band at 1038 cm^−1^ is caused by the stretching mode of Si–O–Si, and the broad absorption band at 1638 cm^−1^ and 1476 cm^−1^ relates to the deformation mode of -NH_2_ and -CH. The strong bands at 2935 cm^−1^ and 2871 cm^−1^ are associated with the stretching mode of -CH_2_. All this confirms the presence of APTES on the surface. For Fe_3_O_4_@CeO_2_-APTES-DFO, the absorption band at 1555 cm^−1^ is caused by the stretching mode of C=O. This confirms the existence of DFO. The strong band at 1413 cm^−1^ is related to the stretching mode of C-O[COOH]. This confirms the presence of HA in Fe_3_O_4_@CeO_2_-APTES-DFO-Zr-HA. Using XPS analysis, the peak of the NH_2_ (primary amine) bond, which is the end of APTES, was observed in Figure 2c, and it was confirmed that the NHCO (secondary amine) bond was produced by combining the amine group of APTES and the carboxyl group of HA in Figure 2d, thus confirming that the surface was coated with HA.

### 3.2. X-ray Diffraction (XRD)

The crystal structure and crystal phase were confirmed by XRD analysis (Figure 3). Fe3O4 and Fe_3_O_4_@CeO_2_ were both analyzed. Diffraction peaks for Fe_3_O_4_ were shown at 30.2°, 35.5°, 43.2°, 57.2° and 62.7°, confirming the lattice planes of (220), (311), (400), (511) and (440) [Joint Committee on Powder Diffraction Standards (JCPDS) No. 65-3107]. The diffraction peaks of CeO_2_ appeared at 28.6°, 33.1°, 47.6° and 56.5°, which were the same peaks as Fe_3_0_4_ in the lattice planes of (111), (200), (220) and (311) (JCPDS No.43-1002). Therefore, the synthesis of Fe_3_O_4_@CeO_2_ was confirmed.

### 3.3. Field Emission Transmission Electron Microscope (FE-TEM)

The Fe_3_O_4_ nanoparticle size was 0.25 nm and appeared on the (311) plane with a 35.5° diffraction peak, while the CeO_2_ was 0.32 nm in size and appeared on the (111) plane with a 28.6° diffraction peak (Figure 4a). This confirms the presence of CeO_2_ on the surface of Fe_3_O_4_. In the case of Fe_3_O_4_@CeO_2_-APTES-HA, the particle size was less than 30 nm, and the coating of HA was observed to be 1 nm thick or greater, confirming that it was successful (Figure 4b,c). An HA coating was attempted on Fe_3_O_4_@CeO_2_, and the result was almost no coating. It can be seen that if APTES is not bound to the surface, applying a coating becomes difficult, because there is no amine group in the terminal group (Figure 4d,e).

### 3.4. Field Emission-Scanning Electron Microscope (FE-SEM) and Energy-Dispersive X-ray Spectroscopy (EDS)

The prepared Fe_3_O_4_, Fe_3_O_4_@CeO_2_ and Fe_3_O_4_@CeO_2_-APTES nanoparticles had diameters of approximately 15 to 30 nm (Figure 5a–c). The Fe_3_O_4_@CeO_2_-APTES-HA particles were about 20 to 30 nm in diameter. Using EDS analysis, 24.1 wt.% of Fe_3_O_4_ Fe was found. In addition, 30.1 wt.% of CeO_2_ Ce and 31.2 wt.% of common O were observed, and 1.8 wt.% of Si in APTES was detected as synthesized on the surface. A large amount of C was coated along with the HA (Figure 5d). The Fe_3_O_4_@CeO_2_-APTES-DFO-Zr had a particle diameter of 15 to 30 nm, and EDS analysis confirmed that a small amount of ^nat^Zr was detected and chelated (Figure 5e). Therefore, the introduction of ^89^Zr, a radioactive isotope, can be expected in the future.

### 3.5. Dynamic Light Scattering (DLS)

According to the DLS analysis results (Figure 6), the particle size of Fe_3_O_4_ was about 171 nm. Fe_3_O_4_@CeO_2_ was about 175 nm. Fe_3_O_4_@CeO_2_-APTES was about 125 nm. Fe_3_O_4_@CeO_2_-APTES-HA was about 146 nm, and Fe_3_O_4_@CeO_2_-APTES-DFO-Zr-HA was about 148 nm. We confirmed that iron oxide tends to agglomerate in an aqueous solution, and it was necessary to improve the dispersion by modifying other substances present on the surface. By examining the polydispersity index (PDI) value, it was confirmed that dispersibility was increased by coating the HA. As a result, fewer side effects due to aggregation in vivo can be expected than those present before coating (Table 1).

### 3.6. Thermogravimetric Analysis (TGA) and Differential Scanning Calorimetry (DSC)

Thermogravimetric analysis was performed to evaluate the effects of the prepared nanoparticles on thermal stability. The gas used was air, the injection rate was 60 mL and the temperature increase rate was 20 °C/min. The measurement range was analyzed using TGA/DSC under the conditions of RT~1100 °C. The mass decreased below 100 °C as a result of moisture, while the Fe_3_O_4_ increased in mass between 100 and 200 °C, and an endothermic reaction occurred, suggesting that the oxygen in the air combined. The endothermic reaction at 600 °C suggests the crystallinity of Fe_3_O_4_. The Fe_3_O_4_@CeO_2_ continued to decrease in mass as a result of moisture, even when above 100 °C, and unlike Fe_3_O_4_, it was calcined in a furnace at 300 °C before the experiment. In Figure 7b, an endothermic reaction can be observed at temperatures above 800 °C, which indicates the crystallinity of CeO_2_. Finally, Fe_3_O_4_@CeO_2_-APTES-DFO-Zr-HA underwent a noticeable mass loss below 500 °C, confirming that the coated HA was affected by temperature. In addition, the graphs in Figure 7 confirm the presence of an HA coating.

### 3.7. In-Vitro Studies: Cytotoxicity

In an in vitro study, the cytotoxicity of the samples was assessed using an MTT assay. In Figure 8a, the cytotoxicity of CeO_2_, according to pH, was evaluated. Both KB and CT-26 showed 30.3% and 26.7% more cytotoxicity when the pH was 6.4 than when the pH was 7.4. This showed that cell growth was inhibited by ROS generation of CeO_2_ under low pH conditions. Figure 8b evaluated target delivery based on the presence of HA. A study on MDA-MB-231 showed that HA-coated Fe_3_O_4_@CeO_2_-APTES-HA was 22.7% more cytotoxic than HA-uncoated Fe_3_O_4_@CeO_2_. Through this experiment, we observed the target transfer capability of HA.

## 4. Conclusions

In this research, we uniformly synthesized core–shell composite nanoparticles with a superparamagnetic core and a ceria shell and successfully coated HA on the surface of the nanoparticles. The synthesis of nanoparticles was confirmed through spectroscopic analysis and thermal analysis, and the coated HA was verified by confirming a clear TEM image. As it was confirmed that the synthesized composite nanoparticles had an average hydraulic diameter of 148 nm, an EPR effect could be expected in the body. In addition, the possibility of introducing ^89^Zr, a radioactive isotope for PET, was confirmed by coordinating Zr using DFO on the nanoparticle surface. In conclusion, we can report the structurally precise synthesis of multifunctional composite nanoparticles. Furthermore, these nanoparticles are expected to contribute to the development of target-directed nano-drug carrier synthesis.

## Figures and Tables

**Figure 1 micromachines-12-01018-f001:**
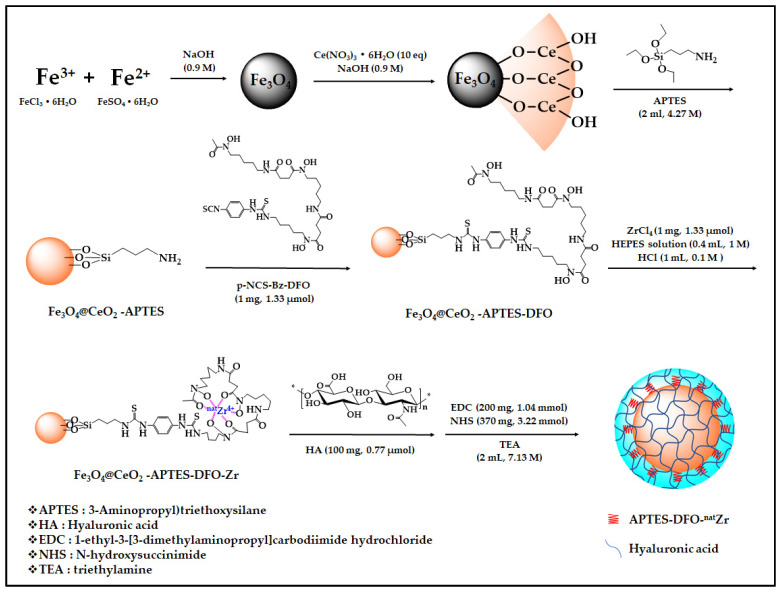
Total scheme of drug delivery system synthesis.

**Figure 2 micromachines-12-01018-f002:**
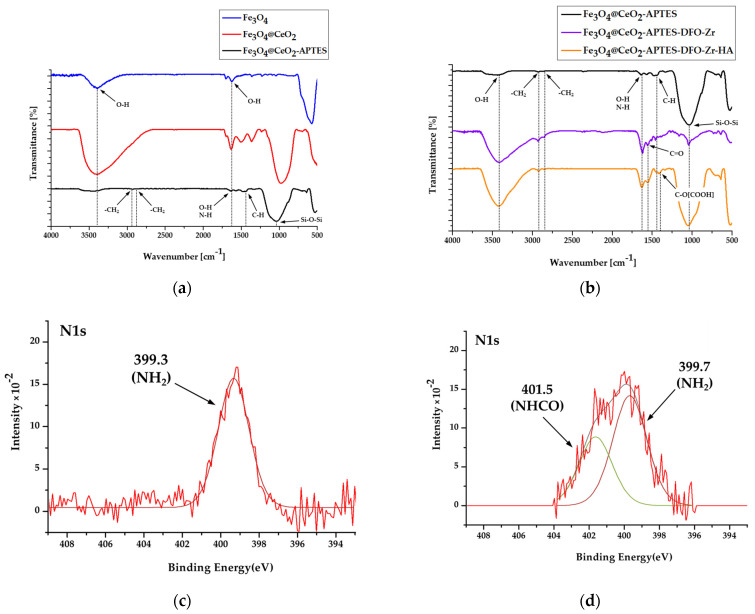
Fourier transform infrared spectroscopy (FT-IR) spectra of: (**a**) Fe_3_O_4_, Fe_3_O_4_@CeO_2_ and Fe_3_O_4_@CeO_2_-APTES; FT-IR spectra of (**b**) Fe_3_O_4_@CeO_2_-APTES, Fe_3_O_4_@CeO_2_-APTES-DFO and Fe_3_O_4_@CeO_2_-APTES-DFO-Zr-HA; X-ray photoelectron spectroscopy (XPS) spectra of (**c**) N1s of Fe_3_O_4_@CeO_2_-APTES; and (**d**) N1s of Fe_3_O_4_@CeO_2_-APTES-DFO-Zr-HA. Acronyms used: (3-Aminopropyl)triethoxysilane (APTES), desferoxamine (DFO).

**Figure 3 micromachines-12-01018-f003:**
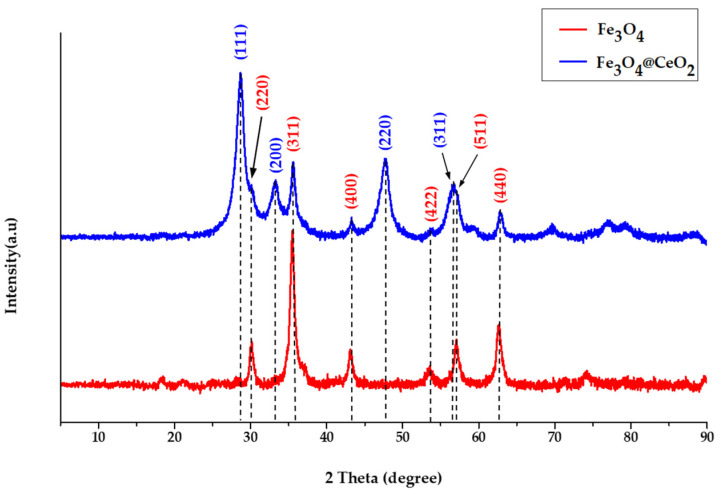
X-ray Diffraction (XRD) patterns of co-precipitated Fe_3_O_4_ and synthesized Fe_3_O_4_@CeO_2_.

**Figure 4 micromachines-12-01018-f004:**
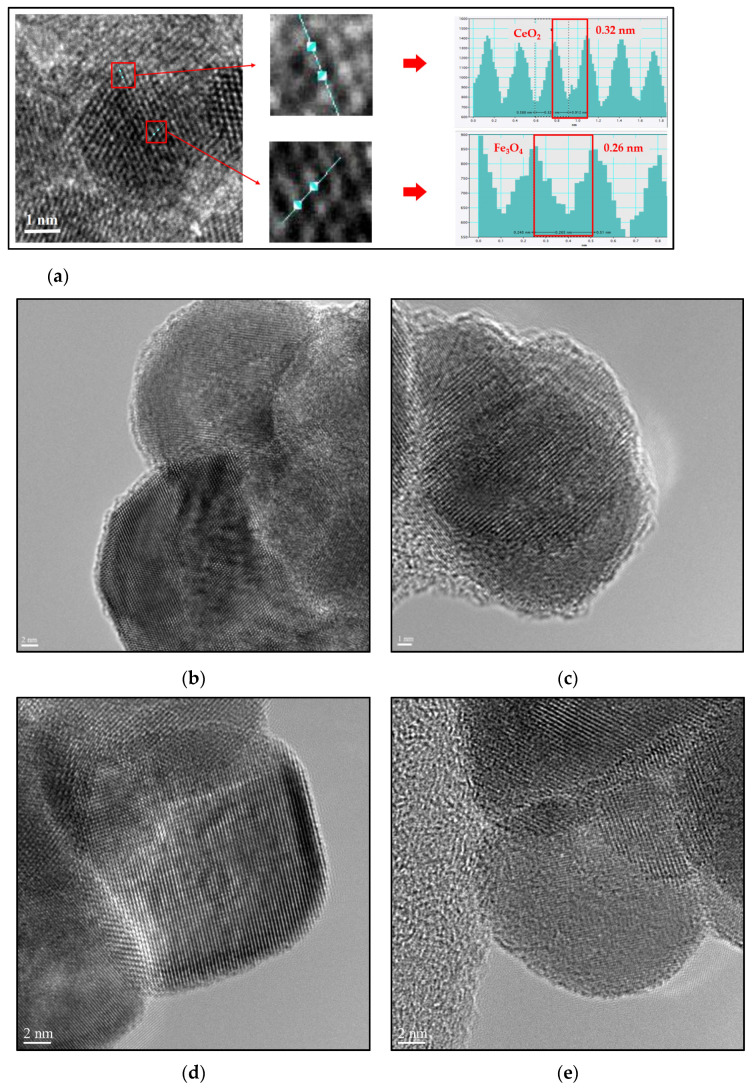
(**a**) Field Emission Transmission Electron Microscope (FE-TEM) images and lattice fringe spacing of Fe_3_O_4_@CeO_2_; (**b**,**c**) FE-TEM images for confirmation of coated HA of Fe_3_O_4_@CeO_2_-APTES-HA nanoparticles and (**d**,**e**) Fe_3_O_4_@CeO_2_ trying to coat hyaluronic acid (HA).

**Figure 5 micromachines-12-01018-f005:**
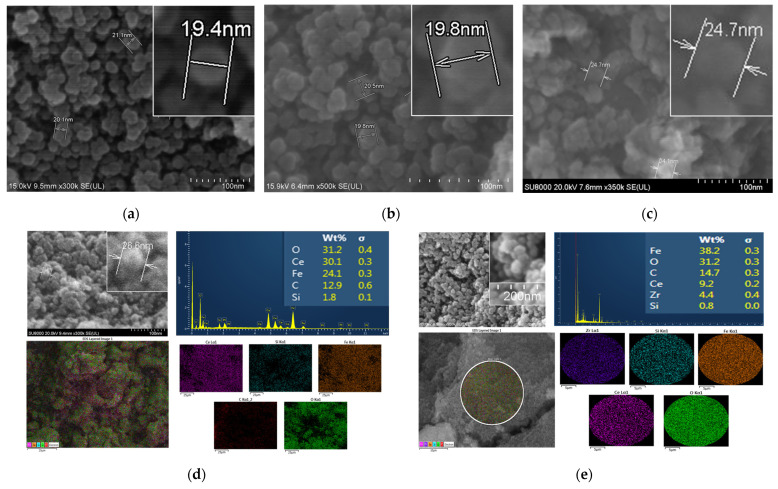
Field Emission-Scanning Electron Microscope (FE-SEM) images for nanoparticle sizing of: (**a**) Fe_3_O_4_, (**b**) Fe_3_O_4_@CeO_2_ and (**c**) Fe_3_O_4_@CeO_2_-APTES; EDS analysis for ingredient identification of: (**d**) Fe_3_O_4_@CeO_2_-APTES-HA and (**e**) Fe_3_O_4_@CeO_2_-APTES- DFO-Zr.

**Figure 6 micromachines-12-01018-f006:**
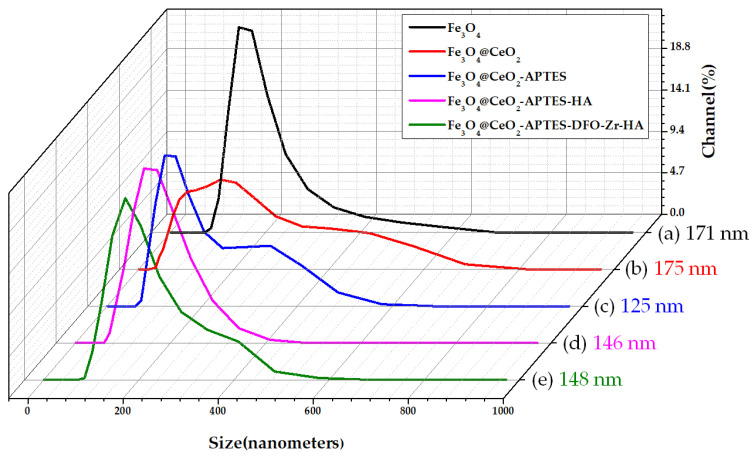
DLS analysis graph for the average particle size determination of: (**a**) Fe_3_O_4_, (**b**) Fe_3_O_4_@CeO_2_, (**c**) Fe_3_O_4_@CeO_2_-APTES, (**d**) Fe_3_O_4_@CeO_2_-APTES-HA and (**e**) Fe_3_O_4_@CeO_2_-APTES-DFO-Zr-HA in dispersed forms.

**Figure 7 micromachines-12-01018-f007:**
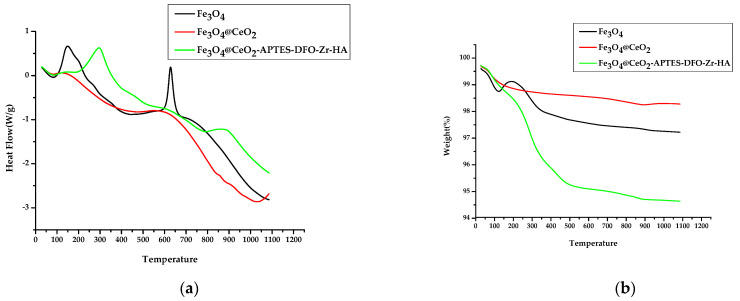
Thermogravimetric Analysis (TGA) measurements for the identification of coated HA of: (**a**) Fe_3_O_4_, Fe_3_O_4_@CeO_2_ and Fe_3_O_4_@CeO_2_-APTES-DFO-Zr-HA; Differential Scanning Calorimetry (DSC) measurements to measure the change in physical properties of substances of (**b**) Fe_3_O_4_, Fe_3_O_4_@CeO_2_ and Fe_3_O_4_@CeO_2_-APTES-DFO-Zr-HA at different temperatures.

**Figure 8 micromachines-12-01018-f008:**
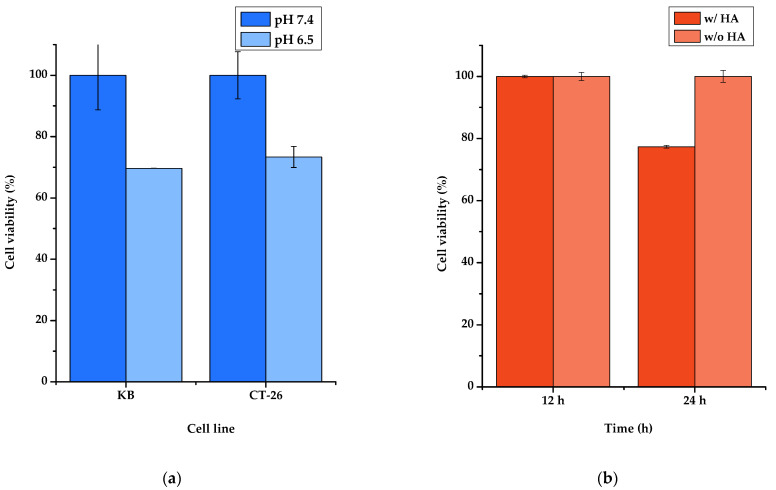
(**a**) Cytotoxicity study using Fe_3_O_4_@CeO_2_ (100 µg/mL) conducted on KB and CT-26 for 12 h; (**b**) cytotoxicity study using Fe_3_O_4_@CeO_2_ (100 µg/mL) without HA and Fe_3_O_4_@CeO_2_-APTES-HA (100 µg/mL) with HA coating conducted on MDA-MB-231 for 12 and 24 h.

**Table 1 micromachines-12-01018-t001:** Dynamic Light Scattering (DLS) analysis table for the average particle size determination and polydispersity index (PDI) of Fe_3_O_4_@CeO_2_, Fe_3_O_4_@CeO_2_-APTES and Fe_3_O_4_@CeO_2_-APTES-HA in dispersed forms.

	Size (nm)	PDI
Fe_3_O_4_@CeO_2_	175	0.448
Fe_3_O_4_@CeO_2_-APTES	125	0.732
Fe_3_O_4_@CeO_2_-APTES-HA	146	0.218

## Data Availability

The data presented in this study are available on request from the corresponding authors.

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
