# Peer review of "Synthesis of Hyaluronic Acid-Conjugated Fe3O4@CeO2 Composite Nanoparticles for a Target-Oriented Multifunctional Drug Delivery System"

_micromachines, 2021, doi:10.3390/mi12091018_

Round 1

Reviewer 1 Report

In this work, "Synthesis of hyaluronic acid-conjugated Fe3O4@CeO2 composite nanoparticles for a target-oriented multifunctional drug delivery system", the authors showed that iron oxide NPs can be utilized towards selective removal of cancer cells. Based on the observed results, the authors claimed that Fe3O4@CeO2 NPs can effectively hinder cancer cells through the activity of CeO2. Overall, this manuscript has a strong potential for a second review after applying the issues and addressing the shortcomings listed below:

1-The authors should polish/revise some grammatical mistakes and typos along the manuscript. I invite the authors to read their manuscript carefully and make the required changes where necessary. 

2-In the Introduction section, while discussing the recent developments in the field of drug delivery and biosensing, the following works should be considered and cited to give a more general view to the possible readers of the work: [(i) Terahertz plasmonics: the rise of toroidal metadevices towards immunobiosensings, Materials Today 32, 108-130 (2020); (ii) Functionalized terahertz plasmonic metasensors: femtomolar-level detection of SARS-CoV-2 spike proteins, 177, 112971 (2021)].

3-In the Introduction section, the information regarding the main focus of this study is missing (especially at the last part of the Introduction). Please add this information.

4-If possible, increase the resolution of Figures 2&6 and use a bit larger arrows in Figure 2. 

5-In terms of consistency, use the same legend style in Figures 2,3, and 6.

6-It seems we have two Figure 3 in the manuscript. Please fix this error and make the required changes along the manuscript. 

7-In Figures 4a,b, increase the size of the text inside the FE-SEM images. Plus, increase size of the overall texts in Figures 4d,e. In their current form, they are hard to read. 

8-In Figure 5, for the pink-, blue-, and black-colored curves: It seems we a flat top (not like the green curve). Is this because of the less data point? Please explain and try to solve the issue. 

Author Response

Dear. Reviewer

Thank you for making a suggestion for my thesis.

I was also able to think a lot after seeing your suggestion, and I could sympathize deeply with your suggestion, so I reviewed it again.

I hope you and your institute will continue to prosper.

Thank you.

Warm regards

Chang Ryong Lee

Response to Reviewer 1 Comments

1-The authors should polish/revise some grammatical mistakes and typos along the manuscript. I invite the authors to read their manuscript carefully and make the required changes where necessary. 

Response 1 : Based on your comments, I have read the premise and revised it again. And I recently submitted a manuscript " Synthesis of Hyaluronic Acid-conjugated Fe3O4@CeO2 Composite Nanoparticles for a Target oriented Multifunctional Drug Delivery System" to MDPI's English editing service

2-In the Introduction section, while discussing the recent developments in the field of drug delivery and biosensing, the following works should be considered and cited to give a more general view to the possible readers of the work: [(i) Terahertz plasmonics: the rise of toroidal metadevices towards immunobiosensings, Materials Today 32, 108-130 (2020); (ii) Functionalized terahertz plasmonic metasensors: femtomolar-level detection of SARS-CoV-2 spike proteins, 177, 112971 (2021)].

Response 2 : I have read the thesis you suggested and I think your opinion is valid, so I have cited the thesis you recommended to make up for the lack of introduction. (Line 35, 353-357)

3-In the Introduction section, the information regarding the main focus of this study is missing (especially at the last part of the Introduction). Please add this information.

Response 3 : Based on your comments, we've added information to complement the main focus. (Line 137-145)

4-If possible, increase the resolution of Figures 2&6 and use a bit larger arrows in Figure 2. 

Response 4 : All pictures have been modified according to your comments. [Figures 2, Figures 7(Figure 6 has been modified to Figure 7.)]

5-In terms of consistency, use the same legend style in Figures 2,3, and 6.

Response 5 : For consistency, I modified them all according to your suggestion. (Figures 2,3, and 7)

6-It seems we have two Figure 3 in the manuscript. Please fix this error and make the required changes along the manuscript. 

Response 6 : Based on your comments, we found an error and corrected it from Figure 3 to Figure 4.

7-In Figures 4a,b, increase the size of the text inside the FE-SEM images. Plus, increase size of the overall texts in Figures 4d,e. In their current form, they are hard to read. 

Response 7 : In accordance with your comments, the size of the internal image of the FE-SEM images has been increased and the size of the entire text has also been modified. (Figures 5)

8-In Figure 5, for the pink-, blue-, and black-colored curves: It seems we a flat top (not like the green curve). Is this because of the less data point? Please explain and try to solve the issue. 

Response 8 : As you said, if you look at the graph again, it seems that there is an error because it is a 3D graph, but it is not an error(As you said, it's also because the data points are so small that they appear that way). So I tabulated the nanoparticles I want to highlight and added their PDI values.(Table 1)

Reviewer 2 Report

In this manuscript, authors have studied the fabrication and post-fabrication of functional nanoparticles for targeted drug delivery systems. This article describes how to fabricate surface modified superparamagnetic iron oxide nanoparticles for a controlled generation of reactive oxygen species (ROS) in tumour microenvironment. Authors have also incorporated hyaluronic acid (HA) coating on the surface of their developed nanoparticles to improve the dispersity as well as  bioactivity. Authors have claimed the fabricated nanoparticles for ROS generation; however, it requires their attention to consider a set of control experiments and confirm their claims in this manuscript. The fabrication and post-fabrication approaches are not convincing the formation of final products due to the lack of appropriate control studies by checking the ROS generation, cell behaviours/responses, and insufficient chemistry analyses, etc. In my view, this manuscript (ID: micromachines-1339611) lacks novelty, and does not provide appropriate and sufficient experimental information and outputs for readers, and not suitable for the publication.

Some general suggestions:  

  1. There are two different FTIR spectra for APTES conjugated nanoparticles (Fe3O4@CeO2-APTES), and both of them should show the presence of amines.
  2. The presence of primary and secondary amines and their differences in this conjugation procedure has not been appreciated in this study, and may use ATR-FTIR for a precise analysis.   
  3. XPS is also required. 
  4. Authors should clarify the HA conjugation, and demonstrate what coupling procedure is leading for the HA conjugation on the nanoparticles. The roles of triethylamine, and EDC/NHS for this conjugation/chemical binding is not clear, and has not proven by means of the applied analyses. For example, the HA polymer are also able to coat the nanoparticles due to the surface charge interactions (the reduced weight as shown by DSC), and not by their designed coupling interactions.
  5. Authors have shown the DLS analysis data, but such important info including PDI, and zeta potential (mV) are missing. It was claimed a size reduction due to lessened aggregation after coating by means of HA, and no PDI data.           
  6. Authors should demonstrate the strong binding of 89Zr, and no release.
  7. Authors should provide minimum required cell culture experiments (ROS generation, cell viability, and targeted internalization, etc.) in order to support their claims throughout the MS. 
  8. Title, abstract and introduction show and discuss many claims (targeted delivery, ROS generation, and EPR effects, etc.); however, nothing has been provided except a couple of chemistry data related to FTIR, DSC, TEM.

Author Response

Dear. Reviewer

First of all, thank you for making a suggestion for my thesis.

I was also able to think a lot after seeing your suggestion, and I could sympathize deeply with your suggestion, so I reviewed it again.

I hope you and your institute will continue to prosper.

Thank you.

Warm regards

Chang Ryong Lee

Response to Reviewer 2 Comments

1.There are two different FTIR spectra for APTES conjugated nanoparticles (Fe3O4@CeO2-APTES), and both of them should show the presence of amines.

Response 1 : Thanks to your comments, I corrected the error of FTIR with APTES(Figure 2a,b).

2.The presence of primary and secondary amines and their differences in this conjugation procedure has not been appreciated in this study, and may use ATR-FTIR for a precise analysis.

Response 2 : I was hoping to use the ATR-FTIR as per your advice, but the analysis was difficult due to time constraints. However, the difference between the primary and secondary amine groups could be explained by analysis through XPS. (Line 229-233, Figure 2c,d)

3.XPS is also required. 

Response 3 : Analysis was performed using XPS as per your advice, and the difference between primary and secondary amine groups can be explained by analysis using XPS. (Lines 229-233, Figure 2c,d)

4.Authors should clarify the HA conjugation, and demonstrate what coupling procedure is leading for the HA conjugation on the nanoparticles. The roles of triethylamine, and EDC/NHS for this conjugation/chemical binding is not clear, and has not proven by means of the applied analyses. For example, the HA polymer are also able to coat the nanoparticles due to the surface charge interactions (the reduced weight as shown by DSC), and not by their designed coupling interactions.

Response 4 : The answer to your comment is that the role of EDC/NHS is an activation approach, in which the amine group of APTES and the carboxyl group of HA help to bind. TEA serves as a buffer solution and creates conditions for good binding by maintaining basicity (Line 196-200). Additionally, among your comments, my opinion on the interaction by design (Figure 4d, e) shows that HA hardly adheres to the surface of nanoparticles when the reaction is performed without APTES. On the other hand, in (Figure 4b,c), it can be compared that HA is well coated due to the amine group of APTES.

5.Authors have shown the DLS analysis data, but such important info including PDI, and zeta potential (mV) are missing. It was claimed a size reduction due to lessened aggregation after coating by means of HA, and no PDI data.    

Response 5 : Following your advice, we measured the PDI values of the nanoparticles we would like to highlight, and found that the agglomeration was reduced after HA coating. (Table 1)

6.Authors should demonstrate the strong binding of 89Zr, and no release.

Response 6 : Based on your advice, we tried to do research using 89Zr, but the time to manufacture 89Zr, a radioactive isotope, is insufficient, so research to support your claim is difficult. However, we used natZr instead of 89Zr and a DFO with proven strong binding based on reference (Line 404-405, 407-412). In the future, based on this research, we can expect to introduce 89Zr, and we plan to conduct small animal experiments.

7.Authors should provide minimum required cell culture experiments (ROS generation, cell viability, and targeted internalization, etc.) in order to support their claims throughout the MS. 

Response 7 : Based on your comments, a cytotoxicity study was conducted to support our claim. Studies have shown that the effect of inhibiting the growth of cells caused by ROS production depends on the pH. (Figure 8a, Line 311-315)

8.Title, abstract and introduction show and discuss many claims (targeted delivery, ROS generation, and EPR effects, etc.); however, nothing has been provided except a couple of chemistry data related to FTIR, DSC, TEM.

Response 8 : Cell experiments were performed according to your comments. The target delivery ability of HA can be confirmed by the characterization of MDA-MB-231 (CD44 overexpressing human breast cancer cells) and the ROS generation of CeO2 by pH-dependent cytotoxicity evaluation. (Figure 8a,b and Line 311-318)

Round 2

Reviewer 1 Report

In its current form, the revised manuscript is suitable for publication.

Reviewer 2 Report

Accept